# 17-β-Estradiol—β-Cyclodextrin Complex as Solid: Synthesis, Structural and Physicochemical Characterization

**DOI:** 10.3390/molecules28093747

**Published:** 2023-04-26

**Authors:** Anna Helena Mazurek, Łukasz Szeleszczuk, Kostas Bethanis, Elias Christoforides, Marta Katarzyna Dudek, Monika Zielińska-Pisklak, Dariusz Maciej Pisklak

**Affiliations:** 1Department of Organic and Physical Chemistry, Faculty of Pharmacy, Medical University of Warsaw, Banacha 1 Str., 02-093 Warsaw, Poland; anna.mazurek@wum.edu.pl (A.H.M.);; 2Doctoral School, Medical University of Warsaw, Żwirki i Wigury 81 Str., 02-093 Warsaw, Poland; 3Laboratory of Physics, Department of Biotechnology, Agricultural University of Athens, 11855 Athens, Greece; 4Structural Studies Department, Centre of Molecular and Macromolecular Studies, Polish Academy of Sciences, Sienkiewicza 112 Str., 90-363 Łódź, Poland; 5Department of Pharmaceutical and Biomaterials Chemistry, Faculty of Pharmacy, Medical University of Warsaw, Banacha 1 Str., 02-093 Warsaw, Poland

**Keywords:** cyclodextrin, estradiol, DFT, SCXRD, solid state NMR

## Abstract

17-β-estradiol (EST) is the most potent form of naturally occurring estrogens; therefore, it has found a wide pharmaceutical application. The major problem associated with the use of EST is its very low water solubility, resulting in poor oral bioavailability. To overcome this drawback, a complexation with cyclodextrins (CD) has been suggested as a solution. In this work, the host–guest inclusion complex between the ß-CD and EST has been prepared using four different methods. The obtained samples have been deeply characterized using ^13^C CP MAS solid state NMR, PXRD, FT-IR, TGA, DSC, and SEM. Using SCXRD, the crystal structure of the complex has been determined, being to the best of our knowledge the first solved crystal structure of an estrogen/CD complex. The periodic DFT calculations of NMR properties using GIPAW were found to be particularly helpful in the analysis of disorder in the solid state and interpretation of experimental NMR results. This work highlights the importance of a combined ssNMR/SCXRD approach to studying the structure of the inclusion complexes formed by cyclodextrins.

## 1. Introduction

17-β-estradiol, EST (Figure 1), is the most potent form of naturally occurring estrogens [1]; therefore, it has found wide application in hormonal contraception, hormone replacement therapy (HRT), and treatment of menopausal and postmenopausal symptoms [2]. Oral administration of EST in a solid dosage form is the most favorable form of HRT [3]. While in the European Pharmacopoeia only the hemihydrate form of EST is described, recently its anhydrous form was successfully obtained [4]. Moreover, numerous cocrystals of EST have been designed [5,6] to solve one of the major problems associated with the application of EST: its poor oral bioavailability caused by very low water solubility—0.2–5 μg mL^−1^ [7].

Cyclodextrins (CDs) are cyclic oligosaccharides consisting of a macrocyclic ring formed by glucose subunits joined by α-1,4 glycosidic bonds. CDs are primarily used in pharmaceutical formulations due to their unique properties, resulting in their ability to form inclusion complexes [8]. The desirable properties of CDs in the pharmaceutical field can be explained at the molecular level. CD molecules resemble a “doughnut” ring, in which small, non-polar substances such as EST can be entrapped. The external fragments of CD molecules are polar due to the presence of hydroxyl groups. When a non-polar substance (e.g., an EST) enters the molecular hole of cyclodextrin, the formed host–guest complex is polar (at outside) and, therefore, is more soluble than the separated guest molecule. Therefore, CDs are commonly used in pharmaceutical formulations as they are able to increase the solubility of APIs, protect them against external factors, such as light, humidity, and heat, or can even mask unpleasant smells or flavors of drugs. Currently, more than 100 original drugs are manufactured with CDs as excipients [9,10,11].

Multiple preparation methods for CD inclusion complexes are being exploited, such as the solvent evaporation method, grinding method, ultrasonic method, and freeze-drying method. It has been shown in many examples that the method of complex preparation may have a major impact on the obtained form of the final product [12,13,14].

Only a small amount of CD complexes have been reported with their crystal structures. This is caused by the fact that many of these complexes are either amorphous or polycrystalline, and even for the crystalline complexes, it is usually very hard to obtain a crystal of a size suitable for single-crystal X-ray measurements [15]. However, in order to fully understand the aforementioned changes resulting from complexation, knowledge of the molecular structure of CD complexes is crucial.

To achieve this goal—that is, to understand the structure and dynamics of the CD-based complexes—multiple computational and analytical methods are usually applied, some of which have been recently reviewed by us [16,17,18]. Among the analytical methods most commonly used to study these kinds of materials in a solid state are Fourier-transform infrared spectroscopy (FT-IR) and powder X-ray diffraction (PXRD), together with thermo-analytical techniques such as differential scanning calorimetry (DSC) and thermogravimetric analysis (TGA). Moreover, the application of solid-state nuclear magnetic resonance (ssNMR) can provide essential information, unobtainable by other methods [16]. From the theoretical approaches, the most important and accurate methods are obtained through the use of quantum chemical calculations, usually at the density functional theory (DFT) level [19]. A recent review [17] revealed that the application of quantum chemical calculations in studies of CD complexes can be essential, providing results unobtainable by any other method, both experimental and computational. In particular, density functional theory (DFT) methods are among the most accurate and most frequently used to model such systems, with the PBE functional being the method of choice when modeling solid state structures. However, to the best of our knowledge, such computations have not been performed on CD inclusion complexes yet.

The properties and structures of the complexes of EST with various CDs have been studied extensively for the last 30 years [20,21,22,23,24,25,26,27]. For example, in a recent work [28], single-crystal X-ray diffraction (SCXRD) results of the EST/ß-CD complex were presented, determining the unit cell and the crystallographic space group of the crystal structure. In this work, only the atomic coordinates of the host molecule were determined, whereas the encapsulated hormone was not possible to be modeled due to its high disorder. Thus, although a 2:1 host:guest stoichiometry of the complex was estimated (based on the residual density calculated by that incomplete model), the structural information provided was limited to the host molecular arrangement in the crystalline state. On the other hand, in the present work, the crystal structure of the EST/ß-CD complex was fully determined and the atomic positions of both host and guest molecules reveal the orientation of the guest in the ß-CD host dimeric cavity and give valuable information about the intermolecular (host–guest and guest–guest) interactions and the arrangement of the full complex units in the crystalline state.

The aim of our work was to obtain the 17-β-estradiol–β-cyclodextrin complex (EST/ß-CD) by means of four different preparation methods. The obtained samples have been extensively characterized by means of various analytical (ssNMR, FT-IR, SCXRD, PXRD, SEM, DSC, TGA) and computational (periodic DFT) methods to extend knowledge of this complex formation and to study how the method of preparation influences the final results.

## 2. Results and Discussion

### 2.1. SCXRD Results

As was mentioned above, a single crystal analysis of the EST/ß-CD complex has been presented in a previous work [28] where only the host and the water oxygen atoms were located via the collected diffraction data using a Mo X-ray source, whereas the encapsulated hormone was not possible to be modeled as the residual electron density appearing within the host cavity was very low (Δ*ρ* ≤ 1 e·Å^3^) due to the high disorder of the included estradiol. Similar to that work, the EST/ß-CD crystal structure presented here was found to belong to the monoclinic system with space group C2, with roughly the same lattice parameters (Table 1). However, in addition to the host and water oxygen atoms, the coordinates of the encapsulated estradiol atoms were successfully determined. This is likely due to the higher diffracted intensities collected by using a Cu *Ka* X-ray source, resulting in significant higher and discrete difference electron density peaks that allowed for the modeling of the disordered guest.

The determined asymmetric unit of the EST/ß-CD crystal structure contains one host, one guest (with s.o.f. of 0.5), and 10.5 water oxygens distributed over 17 sites (no water hydrogens were included). As the complex crystallizes in the C2 space group, the host:guest stoichiometry is 2:1, with two symmetry-related hosts (denoted as hostA and hostA’) forming a classic “head-to-head dimer” stabilized by the well-known intermolecular hydrogen bonds between their secondary hydroxyls. Τhe geometric features of the host molecule are reported in Appendix A, indicating that β-CD, upon complexation with estradiol, adopts the usual torus-like macrocycle shape and the round conformation due to the formation of the commonly observed intramolecular interglucose O3(n)-H···O2(n+1) hydrogen bonds. The encapsulated estradiol molecule is found disordered over two sites (site S1 and site S2, with s.o.f of 0.25 each). Both occupied sites have the same orientation: the guest is accommodated “axially” inside the dimeric β-CD cavity, with the hydroxyl of its A ring protruding from the primary rim of the host β-CD and its D ring being buried into the dimeric hydrophobic cavity (Figure 2a). More specifically, the mean plane of the estradiol aromatic ring system forms an angle of 95.795 (2)° and 82.404 (7)° in the case of the S1 and S2 occupied sites, respectively, with the mean plane of the glucosidic O4n atoms of the hosts. The oxygen atom of the guest’s A ring hydroxyl is located at a distance of 0.638 (5) Å for S1 and 0.428 (3) Å for S2 above the mean plane of the O6n atoms of hostA, whereas the oxygen of the guest’s D ring hydroxyl is found near the O4n atom plane of hostA’ (the distance between the oxygen and the mean plane of the O4n atoms of hostA’ being 0.1642 (12) Å and 0.112 (8) Å for S1 and S2, respectively). The protruding hydroxyl of the guest’s A ring is at hydrogen bond distance from the primary hydroxyls of the host and the protruding hydroxyl of the guest of the consecutive complex unit. In particular, the A ring hydroxyl of the guest occupying the S1 site can be hydrogen bonded with the fully occupied O(61)H, the 40% occupied O(67B)H of the host (1 − x, y, 1 − z), and the guest’s A ring hydroxyl occupying the S2(1 − x, y, 1 − z) site (Appendix A and Figure 2b). However, as the distance between the two consecutive S1 and S1(1 − x, y, 1 − z) sites is just 2.4 Å, the guest is sterically forbidden to occupy the S1 site in two successive complex units. On the other hand, the A ring hydroxyl of the S2 site can be hydrogen bonded with the O(61)H of the host and the guest occupying the S2 site in the successive (1 − x, y, 1 − z) complex unit (Figure 2c). Thus, the arrangement of the encapsulated estradiol in the consequent dimers could be that of S1-S2(1 − x, y, 1 − z) or S2-S2(1 − x, y, 1 − z) but not S1-S1(1 − x, y, 1 − z) (left column of Figure 2d), whereas the S1-S1(x, y, −1 + z), S1-S2(x, y, −1 + z), and S2-S2(x, y, −1 + z) (right column of Figure 2d) arrangement is also possible.

The “head-to-head” β-CD dimers are arranged according to the channel (CH) packing mode (Figure 2) along the c-axis, the distance and the shift between the centroids of two successive dimers being 15.600(4) and 3.0647(7) Å, respectively. The adjacent channels are stacked via bridge water molecules and host–host intermolecular hydrogen bonds (Figure 2e).

### 2.2. Periodic DFT Calculation Results

As described in Section 2.1 (SCXRD results), due to the disorder in the atomic positions of EST in the crystal structure, two significantly different orientations of the neighboring guest molecules exist in the solid state. Therefore, using the experimental crystal structure and choosing the proper guest molecules, we have created two model periodic structures for the DFT calculations. In the first one, the A-rings of neighboring EST molecules are located close to each other (Figure 3). For the purposes of this study, we have named this structure DAAD. This structure can be also found in the Appendix A.

The other possible orientation was named ADAD, as in this one, the D ring of one molecule of EST is always located next to the A ring of the second EST molecule (Figure 4). This structure can also be found in the Appendix A. The unit cell dimensions of both DAAD and ADAD were exactly the same and can be found in Table 2.

It should be noted here that the DFT calculations performed in this study were done directly on the crystal structures, taking into account the periodicity of the studied system and, explicitly, water molecules. In addition, the initial structures for the geometry optimization calculations were taken directly from the SCXRD measurements, without adding any additional atoms. Therefore, the preparation of the structures for the calculations included solely removing chosen guest molecules and, in some cases, the water molecules if they were disordered over two neighboring positions. This resulted in the same stoichiometry of ADAD and DAAD, which enabled direct comparison of the energy of the studied systems. In addition, despite the symmetry found in both ADAD and DAAD, the structures were optimized without any constraints resulting from their corresponding crystal groups, with both of the structures being treated as P1 systems. This has been done purposely to enable the molecules to relax independently.

The results of the calculations (Table 2) show only slight changes in the unit cell dimensions resulting from the geometry optimization. Both the increase (i.e., ‘a’ for ADAD, ‘c’ for both of the structures) as well as the decrease (i.e., ‘a’ for DAAD and ‘b’ for both of the structures) of some unit cell lengths were observed. This is in agreement with the experimental SCXRD results, indicating that the structure was disordered and various guest orientations in the solid complexes exist. Despite similar unit cell dimensions, the optimized structures differed in their energies, indicating that DAAD is the more stable one by approximately 5 kcal/mol. Although no intermolecular interactions between guest molecules were observed in either of the structures, hydrogen bonds between the EST C3 hydroxyl group and primary hydroxyl groups of BCD were observed in ADAD (Figure 4). It should also be noted that after unit cell optimization, the symmetry in ADAD was no longer observed as the optimized ‘a’ and ‘b’ lengths differed significantly.

The next step in the DFT calculations was computation of the NMR chemical shielding constants for the optimized structures of ADAD and DAAD (which can be found in the Appendix A) using the GIPAW method. The isotropic chemical shielding values were then converted into chemical shifts to facilitate peak assignment of the NMR spectra and to compare the differences between the corresponding experimental and theoretical values obtained for the two optimized models.

### 2.3. ^13^C CP MAS Solid State NMR Analysis

As described both in the introduction as well as in Section 4, in this study, the EST/ß-CD complexes were prepared using four different methods (LYS, STAND, MECH, STANDSHORT). Detailed information on how exactly the samples were prepared can be found in Section 4.2.

To explore whether there are any structural differences between the complexes obtained in different ways, we have chosen the ^13^C CP MAS solid state NMR analysis. The application of this method to the study of CD-based complexes has been recently reviewed by us [16]. The spectra of the complexes (LYS, STAND, MECH, STANDSHORT) and reactants (EST, ß-CD) are presented in Figure 5, Figure 6 and Figure 7. The spectra of the EST and ß-CD were recorded to facilitate the observation of changes in the chemical shifts and shapes of signals occurring upon complexation.

At first glance, in the stacked spectra, scaled in a way that the highest peaks have the same intensity (Figure 5), changes in the shape and number of signals originating from the ß-CD are well visible. However, due to the significantly lower molecular mass of guest than host, the intensities of the signals of EST were very low. After increasing the intensity of the peaks (Figure 6), signals from EST have been revealed in all of the spectra of the complexes, with the exception of one obtained using the LYS method. In the LYS spectrum, signals originating from EST carbon atoms are either not visible or, in the best cases, very broad and low, i.e., in the 40–45 ppm region. This indicates that, as anticipated, the LYS method resulted in the amorphization of the sample, which was further confirmed by PXRD analysis (see Section 2.3).

In the discussion below, we will focus on the signals originating from the guest molecule, EST. This is justified for several reasons. First, the signals of EST, both in the non-complexed and in the EST/ß-CD forms, are usually well separated and sharp. Second, the chemical shifts of the EST carbon atoms occur in a wide range, 10–155 ppm, while all the signals from ß-CD can be found in a much wider range, 60–105 ppm. Moreover, the signals from the ß-CD are characterized by much larger FWHM and are highly overlapping; therefore, their analysis would not be possible without the ambiguous deconvolution.

We have started the NMR analysis from the 145–160 ppm range (Figure 7). In this region of the spectra, two overlapping peaks can be observed, with their maxima, respectively, at 152.9 and 154.8 ppm. Initially, after comparison with the spectrum of EST, in which a peak occurs at 152.8, we have assumed that these peaks originate from the complexed (154.42 ppm) and non-complexed (152.9 ppm) EST molecules. However, the intensities of the signals and areas under them were similar, which could mean that only around half of the EST was successfully complexed. Eventually, after careful analysis of the other regions of the spectra, 7.5–15 ppm (Figure 7), we have changed our initial assumptions. In this aliphatic region, the change in the chemical shift of the EST methyl group can be observed. The complexation resulted in the downfield shift of the single peak by c.a. 1 ppm, from 10.67 to 11.71 ppm. Still, even in some (MECH, STAND) of the spectra of complexes, the signal of the non-complexed EST methyl group could be observed. These observations allowed us to draw two conclusions. First, the two peaks in the spectra of complexes, located at 152.9 and 154.8 ppm, originate from the same carbon atom (C3) of the crystallographically nonequivalent EST molecules. This conclusion was also supported by the results of GIPAW calculations (Table 3), as the calculated chemical shifts for the C3 in ADAD and DAAD differ significantly. The other conclusion was that since a different amount of noncomplexed EST could be detected in the analyzed spectra, the yield of the complexation depends on the choice of the preparation method. As the ratio of the intensities of the peak at 10.67 to 11.71 was found to decrease in the order MECH → STAND → STANDSHORT, the yield was also decreasing in the same manner.

Upon complexation, the chemical shifts of some of the EST signals have only slightly changed (i.e., those from C1, C2, C4–C11). These carbon atoms form the A and B ring of EST and do not form any significant intermolecular interactions with other atoms; also, the conformation of these rings is highly rigid. The most apparent changes in the chemical shift values were observed for the signals occurring in the 21–53 ppm region (Figure 8). In the assignment of these signals, the results of GIPAW NMR calculations were found to be particularly useful. Additionally, the changes between the spectra of MECH, STAND, and STANDSHORT observed in the 10.67–11.71 ppm region (Figure 7) were found to be similar to those in the 21–53 ppm region. For example, the C14 signal has a 49.65 ppm shift in the spectrum of EST and 51.39 ppm in the spectrum of EST/ß-CD. In the spectra of MECH and STAND, the low-intensity signal from the noncomplexed EST can be observed, while in the spectrum of STANDSHORT, it is no longer visible. Similar observations were made for C15 signals, occurring at 22.45 and 23.98 ppm in the complexed and non-complexed forms, respectively (Figure 8).

As mentioned above, the results of the GIPAW NMR calculations (Table 3) were found to be in very good agreement with corresponding experimental ones and facilitated proper signal assignment. The obtained differences between the experimental and theoretical values of δ for the complexes were found to be at a level similar to those of EST, not exceeding 4 ppm and, in most cases, below 3 ppm, with an exception for the C7 signal of ADAD. It should be noted, however, that in the spectra of the complexes, the peaks originating from C6, C7, C11, and C12 are overlapping and of a low intensity. This indicates the high level of dynamic disorder in this part of the guest molecule. All four atoms are chemically similar, as they are all secondary and form six-membered rings. Another explanation for this observation can be a dynamic of the C ring of EST. It has been reported previously that the C ring of EST can adopt either a chair or boat conformation, depending on its crystal form or, in the case of a solution, on the solvent [30]. It is therefore possible that the chair–boat conformational dynamics of the EST C ring can occur in the EST/ß-CD complex, which would explain the shape of the signals from C11 and C12. 

The FT-IR spectra (Figure 9) of both ß-CD and EST have been found to be very similar to those reported previously [24]. The observed differences might have been caused by either the method of spectrum registration or the different degree of crystallinity. As in the case of the ^13^C CP MAS NMR results, the FT-IR spectra of the complexes prepared by different methods have been found to be similar. However, there are also some noticeable differences among them. The signal at 3384 cm^−1^ is much narrower in LYS than in other cases. In addition, on the slope of the signal with a maximum at 2925 cm^−1^, the small signals of EST, overlapped by the wide signal of ß-CD, are the most visible in the spectrum of STAND. Moreover, in the fingerprint area (500–850 cm^−1^), the spectrum of LYS is flatter than the spectra of the complexes prepared by other methods. In all of the studied spectra of the complexes, broad signals in the range of 3000–3500 cm^−1^ can be found. These signals originate from multiple hydrogen bonds present in the studied system. These bonds differ in their length, energy, spectroscopic frequency, and intensity, as shown in previous works [31,32,33,34].

Similar to the spectroscopic method results, the PXRD patterns (Figure 10) of the complexes are quite similar, with the exception of LYS. The lack of reflexes in the PXRD pattern of LYS and the characteristic halos indicate that the sample obtained by this method is amorphous. The PXRD patterns of STAND, MECH, and STANDSHORT are similar to the theoretical ones, simulated using the experimental crystal structure of EST/ß-CD. Characteristic reflexes can be found at the 2Θ values of 6.47, 7.25, 9.8, and 11.95 deg. In addition, two groups of signals in the ranges of 14.7–15.7 and 17.5–18.8 can be found both in the theoretical and experimental patterns. In the STAND, MECH, and STANDSHORT group, the pattern of STANDSHORT is slightly different than the other two. For example, in the PXRD pattern of STANDSHORT, there are no signals at 10.75 and 12.535, which are both present in the other two patterns. These reflexes are also present in the pattern of ß-CD, which indicates that in the samples of both STAND and MECH, some ‘free’ crystalline ß-CD can be found.

Comparison of the thermal analysis results (Table 4, Appendix A) revealed additional differences between the complexes obtained using different preparation methods. The differences in mass loss during heating can be explained by the various ratios between the phases (EST:ß-CD:EST/ß-CD) in the analyzed samples. According to the SCXRD results, the total amount of water in EST and EST/ß-CD is 6.20% and 12.95%, respectively. The amount of water in BCD is variable [35], but usually within the 12.5–16% range. The TGA results for EST and ß-CD are in agreement with their corresponding crystal structures, indicating the total water loss in the analyzed temperature range. Lower than theoretically calculated values obtained for the EST/ß-CD complexes may indicate that some of the water molecules in the structure of EST/ß-CD are characterized by lower crystallographic occupancies and that the amount of crystal water in those complexes is variable. Moreover, the presence of non-complexed EST, with 6.20% water content, additionally decreases the anticipated values of water loss during the heating of the complexes. Significantly lower mass loss has been observed for the LYS sample, which can be explained by the final step of this method, lyophilization, which is used to decrease the amount of water in the sample. The DSC analysis of the complexes revealed two endothermic peaks in STANDSHORT and MECH, the first associated with the dehydration and the second with the decomposition of the sample. The higher enthalpy of dehydration was found in the sample with larger water loss, MECH. In the DSC thermograms of STAND and LYS, no clear endothermic peaks were observed.

The surface morphology of EST/ß-CD complexes were assessed by SEM and the images are provided in Figure 11. As shown in Figure 11a, the SEM picture of STANDSHORT demonstrates a crystalline structure of this sample, dominated by cuboid-like crystals with average dimensions of about 100 μm. The SEM picture of MECH in Figure 11b presents an irregularly shaped crystalline structure. This sample is composed of crystals of different size, ranging from a few μm to a few hundred μm. Additionally, the crystals found in this sample are irregularly shaped. Meanwhile, the image of STAND (Figure 11c) shows a lot of large, prism-like crystals, larger even than those found in STANDSHORT, but with more irregular shapes. Finally, the LYS picture (Figure 11d) reveals the amorphous character of this sample, dominated by small particles of irregular shape, which is also in agreement with the PXRD results.

## 3. Conclusions

In this study, the inclusion complex of 17-β-estradiol and β-cyclodextrin has been prepared by four different methods. We have found that the method of complex preparation influences the final composition of the obtained sample as it affects the yield of complexation. However, regardless of the applied method, only one crystal form of the complex has been obtained, with an exception for a method involving lyophilization that resulted in the formation of an amorphous sample.

It should be noted that EST is an API with a long history of application in the treatment of various conditions, such as hormonal contraception, hormone replacement therapy (HRT), and treatment of menopausal and postmenopausal symptoms. ß-CD, on the other hand, is an excipient used in multiple original drugs currently on the worldwide market. It has also been shown that the complexation of EST with this particular cyclodextrin decreased toxicity of the studied hormone [23]. Moreover, the combination of EST and ß-CD was found to improve the bioavailability of the API by increasing its solubility [28].

The crystal structure of the 17-β-estradiol/β-cyclodextrin complex has been obtained for the first time by means of SCXRD. The 2:1 stoichiometry of host:guest has been determined. It has also been found that the highly disordered encapsulated 17-β-estradiol molecule has a unique orientation inside the dimeric host cavity. Using the solved crystal structure, periodic DFT calculations have been conducted to assess the energy differences between the two modeled structures. The GIPAW NMR calculations for the optimized structures facilitated peak assignment in the ^13^C CP MAS NMR spectra. The ssNMR results confirmed that in the crystal structure of the studied complex, two orientations of 17-β-estradiol exist, as for the C3, two signals in the spectra were observed, indicating that only some of the hydroxyl groups of this carbon atom form hydrogen bonds.

SEM and thermal (TGA/DSC) analysis revealed noticeable differences between the complexes obtained using various methods. PXRD analysis confirmed the formation of the complex in each case, with the exception of LYS, as this sample was proven to be amorphous. No major differences in the FT-IR spectra of the complexes obtained by different methods were observed.

The fact that, despite using numerous methods to obtain the studied complex, only one form has been received may indicate that only one stable crystal form is present in normal conditions. This, however, does not exclude the possibility of polymorphism at other temperature or pressure conditions, especially since in the studied structure significant disorder has been detected, which usually indicates the possibility of polymorphic phase transition.

This work highlights the importance of a combined ssNMR/SCXRD approach to studying the structure of the inclusion complexes formed by cyclodextrins, especially those characterized by a high level of structural disorder.

## 4. Materials and Methods

### 4.1. Materials

17-β-estradiol hemihydrate and β-cyclodextrin were purchased from BIOSYNTH; Biosynth AG, Staad, Switzerland) and used as received, without any further purification. For the mechanochemistry (MECH) method, freshly prepared, twice-distilled Milli-Q (Mq) water (Milli-Q water purification system, Millipore Corp., Waltham, MA, USA), with a conductivity of ~1 µS/cm, was used in the grinding process. 

### 4.2. Methods of Complex Preparation

In this study, four different EST/ß-CD complex preparation methods were applied.

The first one was the **STANDARD [STAND]** method, commonly used in cases of CD inclusion complexes, when crystals of sufficient quality for SCXRD measurement are required. The STANDARD method is a slow-cooling crystallization technique: 60 mg of ß-CD was mixed in a flask with 1 mL distilled water and put into 70 °C water for 20 s to obtain a clear solution. Then the contents of the flask were poured into a beaker. In accordance with the molar mass of ß-CD and EST, the respective amount of EST was added to the beaker to maintain the 1:1 molar ratio. The beaker was put on a magnetic stirrer and left at room temperature for 15–20 min until a clear solution was obtained. Afterwards, the contents of the beaker were poured into a glass tube. The beaker was poured along with 0.5–1.0 mL water, which was also added to the glass tube. The glass tube was held in 70 °C water for 20 s to obtain a clear solution. Later, the tube was closed and put into a 70 °C water bath. A slow, gradual cooling process was performed over 10 days, reaching a temperature of 24 °C on the 10th day. At the end, a rotary evaporator was used.

The second one was the **STANDARD SHORT [STANDSHORT]** method: ß-CD and EST in a molar ratio of 1:1 were mixed with distilled water and put into a round bottom flask. The flask was closed and left on a magnetic stirrer for 24 h at a temperature of 44 °C. To obtain crystals, a rotary evaporator was used.

The third one was the **MECHANOCHEMICAL [MECH]** method: ß-CD and EST in a molar ratio of 1:1 were mixed in a mortar with 3–5 drops of Mq water and knitted for 5 min every day over 5 consecutive days.

The fourth one was the **LYOPHILIZATION [LYS]** method: the **STANDARD SHORT** crystallization method with the application of lyophilization instead of slow evaporation. A solution, which was obtained in accordance with the **STANDSHORT** method, was poured into smaller containers in which the lyophilization process took place. Firstly, the probes were frozen with the application of liquid nitrogen. Secondly, the probes were put into a lyophilizator for 48 h.

### 4.3. Powder X-ray Diffraction (PXRD)

For the PXRD measurements, a Panalytical Empyrean (Malvern, UK) diffractometer was used. The samples were analyzed in Bragg–Brentano reflection mode, using Cu-Kα radiation (λ = 1.54187 Å), a 2Θ range of 3–45°, and a 0.006565° step size. For the incident beam a fixed divergence slit of 1/16°, an anti-scatter slit of 1/4°, and a fixed mask of 4 mm were used, and the diffracted beam path was equipped with a 7.5 mm anti-scatter slit.

### 4.4. Single-Crystal X-ray Diffraction (SCXRD)

A clear, light, colorless prism-like specimen, with dimensions of about 0.130 mm × 0.270 mm × 0.400 mm and coated with paraffin oil as cryo-protectant, was used for the X-ray crystallographic analysis. The X-ray intensity data were measured at 100 (2) K with a Bruker (Billerica, MA, USA) D8-VENTURE diffractometer using Cu *Kα* radiation (λ = 1.54178 Å). A low-temperature device (Oxford Cryosystems Ltd., Long Handorough, UK) provided a continuous stream of nitrogen vapor on the specimen during the data collection, while diffraction patterns were recorded using a CMOS-PHOTON III detector.

The total exposure time was 29.33 h. The frames were integrated with the Bruker SAINT software package [36] using a narrow-frame algorithm. The integration of the data using a monoclinic unit cell yielded a total of 133,843 reflections to a maximum θ angle of 76.27 (0.79 Å resolution). The final cell constants of *a* = 19.1245(15) Å, *b* = 24.4180(18) Å, *c* = 15.6004(11) Å, β = 109.500(5)°, and volume = 6867.2(8) Å^3^ were based upon the refinement of the XYZ-centroids of 9073 reflections above 20 σ(I) with 6.094 < 2θ < 148.5°. Data were corrected for absorption effects using the Multi-Scan method (SADABS) [36].

The structure was solved by the intrinsic phasing method with SHELXT [37] and refined by full-matrix least squares against *F*^2^ using SHELXL-2014/7 [38] through the SHELXLE GUI [39]. H-atoms were placed geometrically and refined in riding mode with isotropic displacement parameters fixed by SHELXL. Due to the structural complexity and disorder of the final model, soft restraints on bond lengths and angles, generated from the PRODRG2 webserver [40], were applied on the host and guest molecules of the inclusion complexes. Anisotropic displacement parameters were refined using soft restraints (SIMU) [41] implemented in the SHELXL program, where necessary. The final anisotropic full-matrix least-squares refinement on *F*^2^ with 950 variables converged at R_1_ = 9.5% for the observed data and wR_2_ = 26.9% for all data. The goodness of fit (GoF) was 1.04. The crystallographic data along with the structural refinement details are summarized in Table 1. The data can be obtained from the Cambridge Crystallographic Data Centre under the reference number 2250781.

Geometric features of the crystal structure, e.g., interatomic distances, angles, dihedral angles, centroid coordinates, and mean plane equations through various groups of atoms, along with their e.s.d. estimations, were calculated via the full covariance matrix using the Olex2 program [42]. The final 3D model was drawn with Mercury 4.3.1 [43] and PyMoL [44].

### 4.5. Fourier-Transform Infrared Spectroscopy (FT-IR)

The studies were performed using a Perkin-Elmer Spectrum 1000 FT-IR spectrometer equipped with an MTEC 300 detector (MTEC, Ames, IA, USA). The samples were packed in ring cups with a diameter of 10 mm. The detector’s chamber was purged with helium to reduce the effect of moisture evaporating from the samples during measurement. A spectrum obtained from the background sample was subtracted from the spectrum of each sample to eliminate the residual peaks of CO_2_ and moisture. For each sample, 1024 scans were recorded and averaged in the infrared region between 4000 and 500 cm^−1^ at a resolution of 4 cm^−1^. All spectral plots were prepared using the GRAMS/AI 8.0 Spectroscopy Software.

### 4.6. Cryo-Scanning Electron Microscopy (Cryo-SEM)

The analysis was performed using low-temperature scanning electron microscope ZEISS AURIGA (Warsaw, Poland) 60 coupled with a focused ion beam. Cryo-SEM allows sample observation without chemical fixing or drying. The procedure consists of sample freezing by immersion in liquid nitrogen, breaking, and etching.

### 4.7. Differential Scanning Calorimetry and Thermogravimetry Analysis (DSC-TGA)

Analysis was performed using apparat SDT Q600 (TA Instruments, New Castle, DE, USA) under nitrogen flow. The heating rate was equal to 10 °C/min and the sample mass was approximately 6–8 mg. Pierced aluminum sample pans were used in the analysis and the temperature range was set to 0–500 °C.

### 4.8. ^13^C CP MAS NMR

Solid-state ^13^C CP/MAS NMR spectra were recorded on a Bruker Avance III 600 spectrometer (Bruker BioSpin, Rheinstetten, Germany) operating at 600.15 MHz (^1^H) and 150.91 MHz (^13^C), and powder samples were spun at 12 kHz in a 4 mm ZrO_2_ rotor using a double air-bearing probe head. Acquisition was performed with a standard CP pulse sequence with ramped CP scheme, 2 ms CP contact time, 4 s recycle delay, and a swept-frequency two-pulse phase modulation decoupling scheme, using a 3.2 μs proton 90° pulse. The decoupling field strength was set to 78 kHz. A total of 256 scans were acquired, 10.00 exponential apodization, a receiver gain equal to 2050, and 2048 acquired points. After zero filling and LP application, the spectrum size was 4096. ^13^C chemical shifts were referenced to adamantane CH_2_ at 38.48 ppm.

### 4.9. Periodic DFT Calculations

The density functional theory (DFT) calculations of geometry optimization and NMR parameters, under periodic boundary conditions, were carried out with the CASTEP program [45] implemented in the Materials Studio 2020 software [46] using the plane wave pseudopotential formalism. On-the-fly-generated ultrasoft pseudopotentials were generated using a Koelling–Harmon scalar relativistic approach [47]. The Perdew–Burke–Ernzerhof (PBE) [48] exchange-correlation functional, defined within the generalized gradient approximation, with Tkatchenko–Scheffler (TS) [49] dispersion correction, was used in the calculations.

#### 4.9.1. Geometry Optimization

Geometry optimization was carried out using the limited memory Broyden–Fletcher–Goldfarb–Shanno (LBFGS) [50] optimization scheme and smart method for finite basis set correction. The kinetic energy cutoff for the plane waves (E_cut_) was set to 630.0 eV. The number of Monkhorst–Pack k-points during sampling for a primitive cell Brillouin zone integration [51] was set to 2 × 2 × 1 (for EST-ß-CD) and 1 × 1 × 2 (for 17-β-estradiol hemihydrate, refcode ESTDOL10), respectively.

The experimental X-ray structure of EST/ß-CD was used to create two initial periodic structures for calculations, containing two EST and four ß-CD molecules in the unit cell each. Details on the structural preparation can be found in Section 2.2.

During geometry optimization, all atom positions and cell parameters were optimized, with no constraints. The convergence criteria were set at 1 × 10^−5^ eV/atom for the energy, 3 × 10^−2^ eV/Å for the interatomic forces, 5 × 10^−2^ GPa for the stresses, and 1 × 10^−3^ Å for the maximum displacement. A fixed-basis set quality method for the cell optimization calculations and a 1 × 10^−6^ eV/atom tolerance for SCF were used.

#### 4.9.2. NMR Parameter Calculations

The computation of shielding tensors was performed using the Gauge Including Projector Augmented Wave Density Functional Theory (GIPAW) method of Pickard et al. [52]. To compare the theoretical and experimental data, the calculated chemical shielding constants (σiso) were converted to chemical shifts (δiso) using the following equation: δiso = (σGly + δGly)—σiso, where σGly and δGly stand for the shielding constant and the experimental chemical shift, respectively, of the glycine carbonyl carbon atom (176.50 ppm).

## Figures and Tables

**Figure 1 molecules-28-03747-f001:**
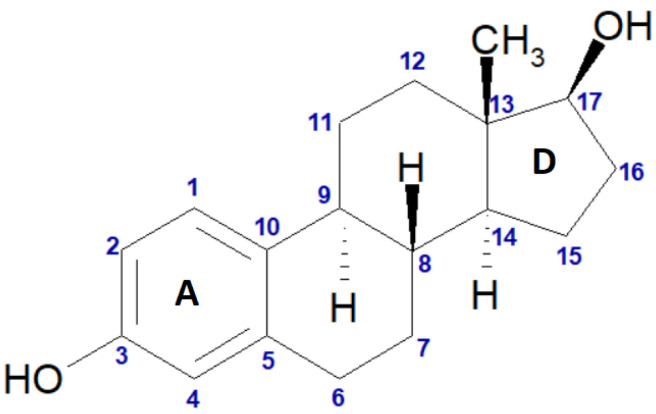
Chemical structure of 17-β-estradiol (EST) with atom numbering. “A” and “D” represent the symbols of the particular rings within the structure.

**Figure 2 molecules-28-03747-f002:**
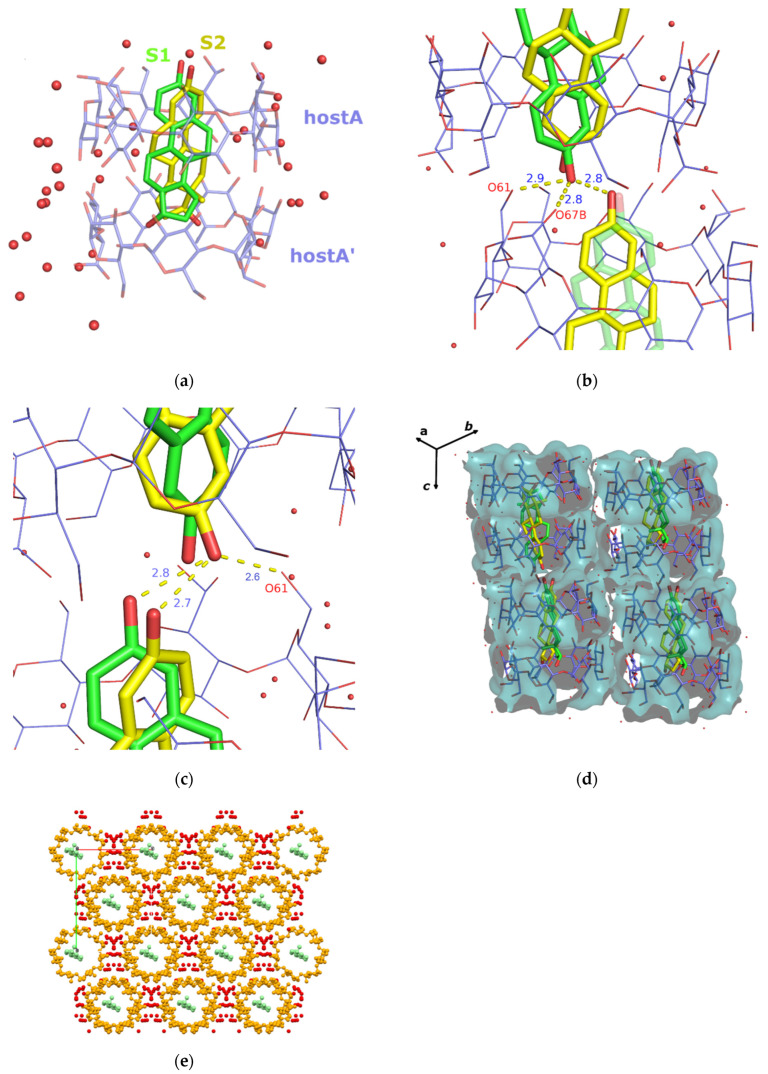
(**a**) Estradiol disordered over two sites (S1 green and S2 yellow) is fully encapsulated inside the cavity of a classic “head-to-head” β-CD host dimer with a 2:1 host:guest stoichiometry. (**b**) Hydrogen bonds between the protruding oxygen atom of the A ring hydroxyl of the guest occupying the S1 site (green) and primary host–guest (occupying the S2 site) of the consecutive complex unit. (**c**) Hydrogen bonds between the guest occupying the S2 site (yellow) and the host–guest (both occupied sites) of the consecutive complex unit. (**d**) β-CD dimers forming channels along the c-axis. Left-hand side: S1-S2(1 − x, y, 1 − z) and S2-S2(1 − x, y, 1 − z) possible arrangements of the encapsulated estradiol molecules in the channel. Right-hand side: S1-S1(x, y, −1 + z), S1-S2(x, y, −1 + z), and S2-S2(x, y, −1 + z) possible arrangements of the encapsulated estradiol molecules in the channel. (**e**) Crystal packing of the EST/β-CD complex view perpendicular to the ab plane. In all figures, hydrogen atoms are omitted for clarity.

**Figure 3 molecules-28-03747-f003:**
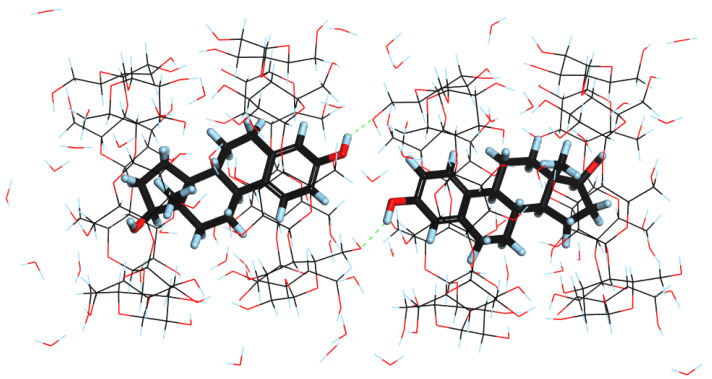
DFT-optimized structure of DAAD. Hydrogen bonds formed between EST and BCD are indicated as green dashed lines.

**Figure 4 molecules-28-03747-f004:**
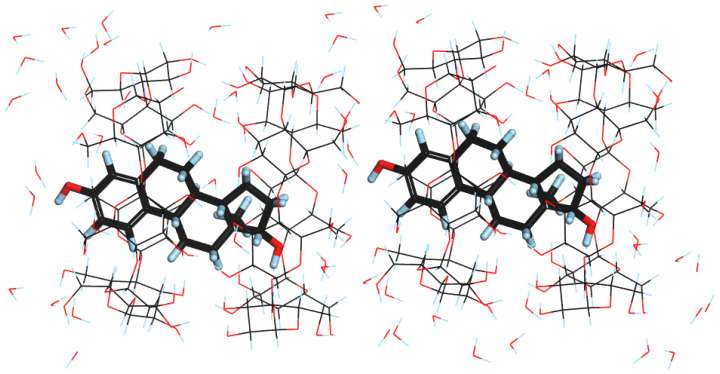
DFT-optimized structure of ADAD.

**Figure 5 molecules-28-03747-f005:**
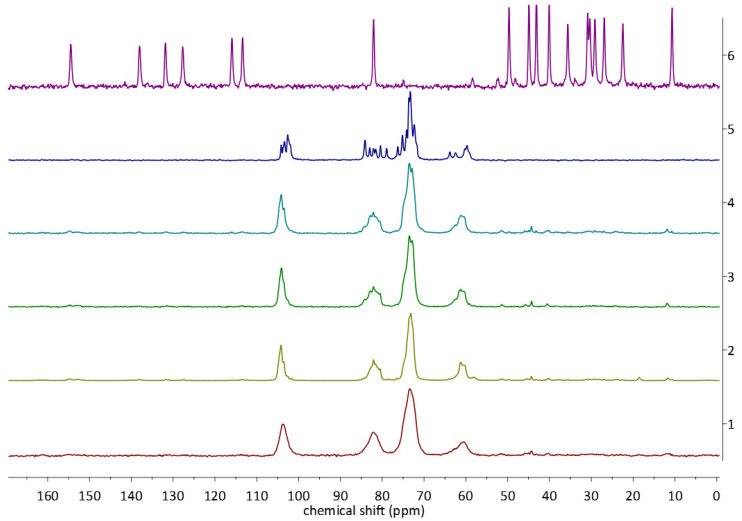
^13^C CP MAS NMR spectra of the EST (violet), ß-CD (dark blue), STAND (blue), STANDSHORT (green), MECH (olive green), and LYS (red).

**Figure 6 molecules-28-03747-f006:**
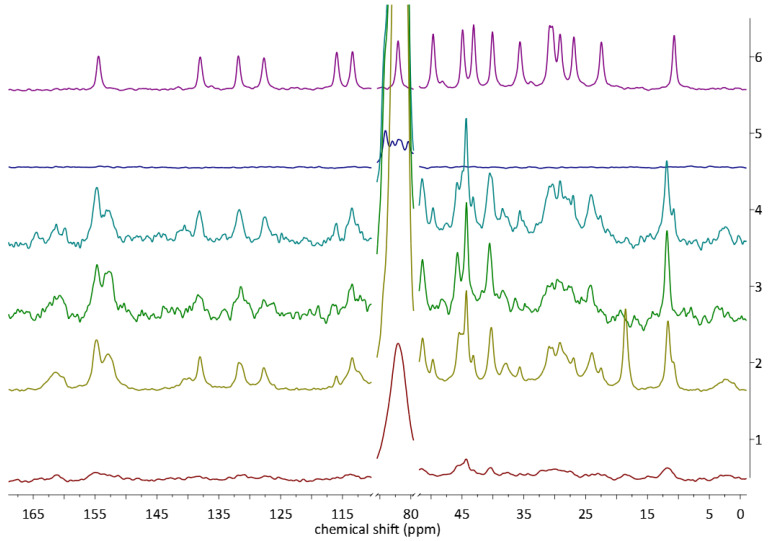
^13^C CP MAS NMR spectra of the EST (violet), ß-CD (dark blue), STAND (blue), STANDSHORT (green), MECH (olive green), and LYS (red). Only chosen regions of the spectra are presented for better visualization of the signals originating from EST.

**Figure 7 molecules-28-03747-f007:**
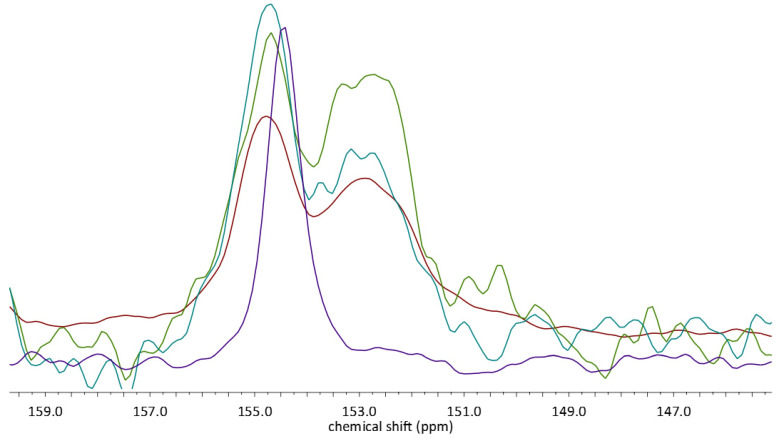
^13^C CP MAS NMR spectra of the EST (violet), STAND (blue), STANDSHORT (green), and MECH (red). Only chosen regions of the spectra are presented for better visualization of the signals originating from EST.

**Figure 8 molecules-28-03747-f008:**
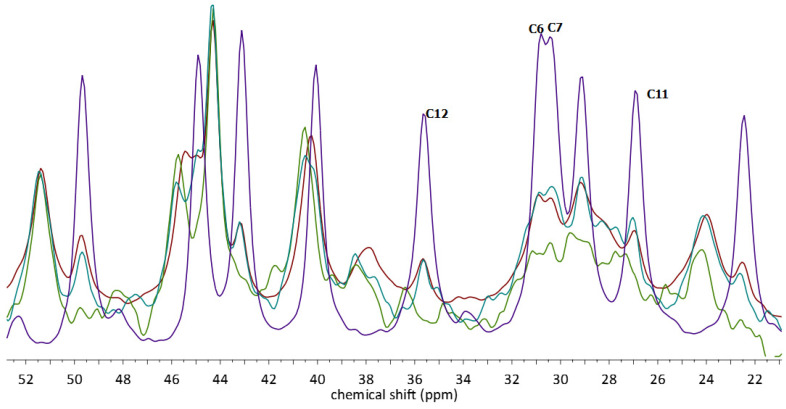
^13^C CP MAS NMR spectra of the EST (violet), STAND (blue), STANDSHORT (green), and MECH (red). Only chosen regions of the spectra are presented for better visualization of the signals originating from EST.

**Figure 9 molecules-28-03747-f009:**
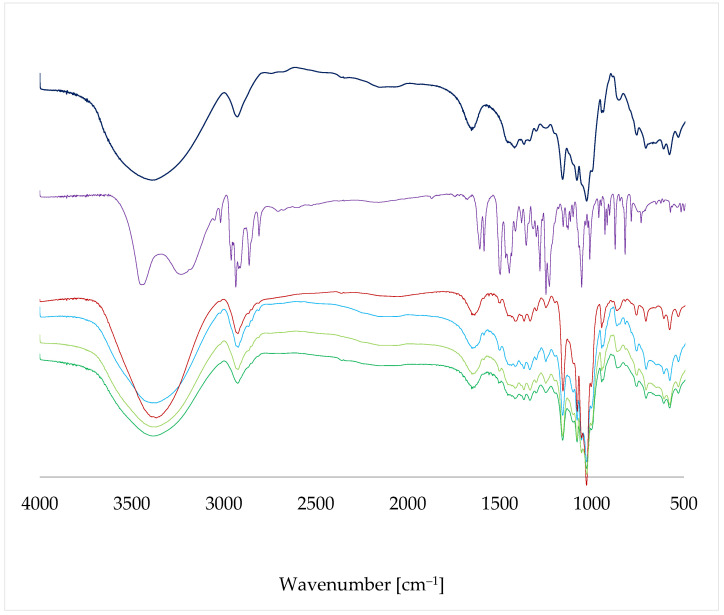
FT-IR spectra of ß-CD (dark blue), EST (violet), LYS (red), STAND (blue), MECH (olive green), and STANDSHORT (green).

**Figure 10 molecules-28-03747-f010:**
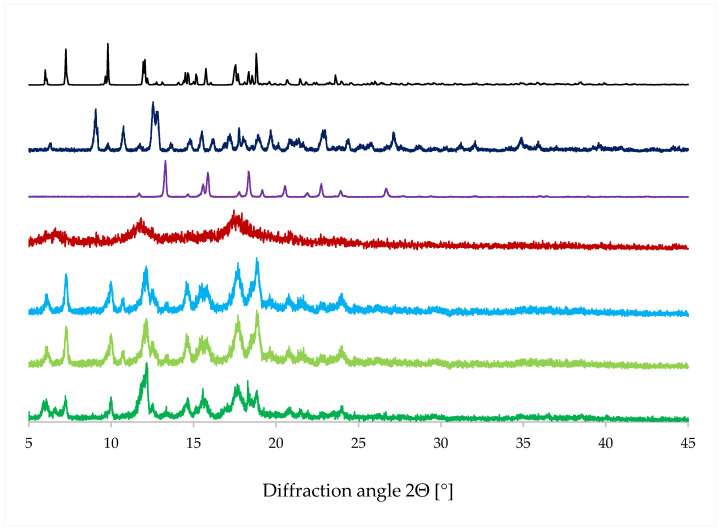
PXRD patterns: simulated for EST/ß-CD (black) and experimental PXRD prints for ß-CD (dark blue), EST (violet), LYS (red), STAND (blue), MECH (olive green), and STANDSHORT (green).

**Figure 11 molecules-28-03747-f011:**
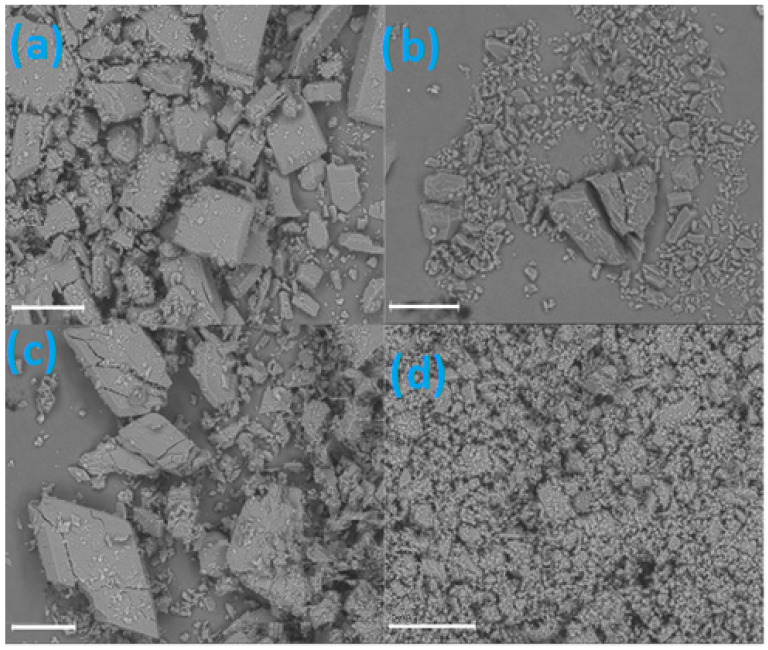
SEM images of STANDSHORT (**a**), MECH (**b**), STAND (**c**), and LYS (**d**). The length of the scale (white horizontal line) in each case is the same: 200 μm.

**Table 1 molecules-28-03747-t001:** Crystallographic parameters of the EST/ß-CD inclusion complex [29].

Crystal Data	EST/ß-CD
CCDC No.	2250781
Complex formula in the asymmetric unit	(C_42_H_70_O_35_)·0.5(C_18_H_24_O_2_)·10.5H_2_O
Formula weight	1439.16
Crystal system, space group	Monoclinic, *C*2
Temperature (K)	100
*a*, *b*, *c* (Å)	19.1245 (15), 24.4180 (18), 15.6004 (11)
α, β, γ (°)	109.500 (5)
V (Å^3^)	6867.2 (9)
Z	4
Radiation type	Cu *Ka*
μ (mm^−1^)	1.09
Crystal size (mm^3^)	0.4 × 0.27 × 0.13
**Data collection**	
*T*_min_, *T*_max_	0.593, 0.754
No. of measured, independent, and observed [I > 2σ(I)] reflections	105,516, 11,945, 10,450
R_int_	0.074
(sin θ/λ)_max_ (Å^−1^)	0.595
**Refinement**	
R_1_ [*F*^2^ > 2σ(*F*^2^)], *w*R_2_(*F*^2^), GooF	0.095, 0.269, 1.04
No. of reflections	11,945
No. of parameters	950
No. of restraints	202
Δ*ρ*_max_, Δ*ρ*_min_ (e Å^−3^)	0.78, −0.44

**Table 2 molecules-28-03747-t002:** Unit cell parameters and relative energy for the modeled structures, ADAD and DAAD, compared with the initial experimental structure used for geometry optimization (Exp.).

	Exp.	ADAD	DAAD
a [Å]	15.515011	15.742737	15.484059
b [Å]	15.515011	15.314397	15.484028
c [Å]	31.188400	31.718429	31.699391
α [°]	101.86701	100.93157	101.31461
β [°]	101.86701	99.83545	101.31466
γ [°]	103.84663	104.42976	101.44811
Relative energy [kcal/mol]		0	−5.152

**Table 3 molecules-28-03747-t003:** Experimental (exp.) and theoretically calculated (GIPAW) ^13^C chemical shifts (δ) of EST and its complex with β-CD. Due to the presence of two EST molecules in the asymmetric unit of both ADAD and DAAD, two sets of values, (1) and (2), have been obtained for the first (1) and second (2) molecule present in the unit cell.

Atom Number	δ EST Exp.	δ EST GIPAW	δ EST Exp.—δ EST GIPAW	δ EST + ß-CD Exp.	δ ADAD GIPAW (1)	δ ADAD GIPAW (2)	(EST + ß-CD Exp.)—ADAD GIPAW (1)	(EST + βCD Exp.)—ADAD GIPAW (2)	δ DAAD GIPAW (1)	δ DAAD GIPAW (2)	(EST + ß-CD Exp.)—DAAD GIPAW (1)	(EST + ß-CD Exp.)—DAAD GIPAW (2)	δ EST Exp.—(EST/ß-CD Exp.)
1	127.65	129.59	−1.94	127.78	127.77	127.76	0.01	0.02	124.65	124.71	3.13	3.07	−0.13
2	113.33	111.18	2.15	113.43	109.86	109.82	3.57	3.61	109.61	109.65	3.82	3.78	−0.1
3	154.42	156.06	−1.64	154.8/152.9	157.21	157.23	−2.41	−2.43	155.72	155.71	−2.92	−2.91	1.62
4	115.89	118.35	−2.46	115.97	113.2	113.2	2.77	2.77	113.24	113.27	2.73	2.7	−0.08
5	137.94	139.1	−1.16	137.98	140.12	139.98	−2.14	−2.00	139.55	139.65	−1.57	−1.67	−0.04
6	30.83	30.75	0.08	30.92	28.79	28.81	2.13	2.11	28.58	28.56	2.34	2.36	−0.09
7	30.36	29.98	0.38	30.36	25.09	25.09	5.27	5.27	26.49	26.49	3.87	3.87	0.00
8	40.05	38.54	1.51	40.31	38	37.94	2.31	2.37	37.13	37.12	3.18	3.19	−0.26
9	44.89	43.52	1.37	45.5	45.24	45.2	0.26	0.30	43.73	43.74	1.77	1.76	−0.61
10	131.77	133.53	−1.76	131.84	132.39	132.31	−0.55	−0.47	130.46	130.52	1.38	1.32	−0.07
11	26.9	25.78	1.12	26.95	26.27	26.29	0.68	0.66	23.63	23.61	3.32	3.34	−0.05
12	35.59	36.07	−0.48	37.76	35.08	35.15	2.68	2.61	35.84	35.84	1.92	1.92	−2.17
13	43.09	42.62	0.47	44.3	43.5	43.38	0.8	0.92	42.08	42.07	2.22	2.23	−1.21
14	49.65	48.09	1.56	51.39	51.23	51.21	0.16	0.18	50.06	50.04	1.33	1.35	−1.74
15	22.45	21.06	1.39	23.98	20.61	20.61	3.37	3.37	21.45	21.43	2.53	2.55	−1.53
16	29.12	28.09	1.03	29.18	26.62	26.61	2.56	2.57	29.53	29.52	−0.35	−0.34	−0.06
17	82.04	82.63	−0.59	82.06	84.46	84.38	−2.4	−2.32	85.32	85.28	−3.26	−3.22	−0.02
18	10.67	7.54	3.13	11.71	8.46	8.5	3.25	3.21	8.54	8.52	3.17	3.19	−1.04

**Table 4 molecules-28-03747-t004:** DSC-TGA analysis results.

**STAND**	DSC	(1) Onset temp. 29.26 °C	(3) Onset temp. 152.0 °C
		Peak temp. 61.83 °C	Peak temp. 177.96 °C
		Enthalpy 83.82 J/g	Enthalpy 10.74 J/g
		(2) Onset temp. 101.1 °C	(4) Onset temp. 202.0 °C
		Peak temp. 111.27 °C	Peak temp. 212.69 °C
		Enthalpy 18.53 J/g	Enthalpy 6.434 J/g
	TGA	Temp. range of dehydration: 27–200 °C	
		Associated mass loss: 6.911%	
**STANDSHORT**	DSC	Onset temp. 26.38 °C	
		Peak temp. 45.33 °C	
		Enthalpy 185.1 J/g	
	TGA	Temp. range of dehydration: 25–200 °C	
		Associated mass loss: 7.034%	
**MECH**	DSC	Onset temp. 30.55 °C	
		Peak temp. 69.66 °C	
		Enthalpy 209.8 J/g	
	TGA	Temp. range of dehydration: 20–200 °C	
		Associated mass loss: 9.408%	
**LYS**	DSC	Onset temp. 41.0 °C	
		Peak temp. 55.01 °C	
		Enthalpy 47.53 J/g	
	TGA	Temp. range of dehydration: 25–200 °C	
		Associated mass loss: 4.065%	
**EST**	DSC	(1) Onset temp. 81.88 °C	
		Peak temp. 104.81 °C	
		Enthalpy 19.70 J/g	
		(2) Onset temp. 175.91 °C	
		Peak temp. 178.20 °C	
		Enthalpy 91.06 J/g	
	TGA	Temp. range of dehydration: 20–200 °C	
		Associated mass loss: 6.20%	
**ß-CD**	DSC	Onset temp. 56.31 °C	
		Peak temp. 91.48 °C	
		Enthalpy 380.8 J/g	
	TGA	Temp. range of dehydration: 20–100 °C	
		Associated mass loss: 12.95%	

## Data Availability

Data can be obtained from the corresponding author (Ł.S.) by email. Crystallographic data have been deposited with the Cambridge Structural Database (CSD) under deposition number CCDC: 2250781.

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
