# Peer review of "17-β-Estradiol—β-Cyclodextrin Complex as Solid: Synthesis, Structural and Physicochemical Characterization"

_molecules, 2023, doi:10.3390/molecules28093747_

Round 1

Reviewer 1 Report

Please see my comments in the attached pdf file,

The English language employed in the manuscript is generally well-written, but there are a few minor grammatical errors that require correction. I suggest proofreading the revised version thoroughly.

Reviewer 2 Report

The manuscript from Mazurek et al. is an interesting work describing a characterization of 17-β-Estradiol incorporated into beta-Cyclodextrin. The work is overall well done, combining together many instrumental techniques and a good combination of the solid-state NMR spectra and DFT chemical shift calculation.

Overall, the manuscript is well written, the conclusions are properly supported and it will interest the chemical community, so I suggest publication on Molecules.

Nevertheless, I have some minor corrections to be done prior publications and reported below.

Minor corrections:

1)    Figures with solid-state spectra have to be improved, spectra should be plotted with thicker lines and axis labels are too small and unreadable.

2)    Beta-Cyclodextrin is reported in the text written in several ways, BCD, b-CD, beta-CD…please use a single mode, I suggest ß-CD.

3)    Calculated chemical shifts in Table 4 are calculated in two different ways, (1) and (2)! These two calculations MUST be better described either in the text, explained in the caption and in the Method part.

4)    In Figure 11: Please put letters A), B), C), and D) in the figure panels and write the scale labels of each panel in a clear way.

5)    The NMR part in the Methods section is too poor and must be better detailed with all the experimental conditions: for example CP contact time and powers, decoupling sequences and power, number of scans, acquired points, processing parameters…….and many other important details.

Reviewer 3 Report

In the present manuscript (molecules-2348482), entitled 17-β-Estradiol - β-Cyclodextrin Complex as Solid: Synthesis,Structural and Physicochemical Characterization, information is presented on a compound widely used in drug biosynthesis. The authors made some experimental and computational works on " Cyclodextrin derivative", which has already very well known. In that work, the authors have used the Materials Studio 2020 software to perform the geometry optimization and some others physicochemical proprieties. The manuscript has potential but needs modest improvement before publication. My comments are attached below.

1-      The novelty of the work should be established.

2-      Qualitative informations are missing in abstract. Abstract should be concise and the authors need to improve with more specific short results.

3-      Why particularly this computational technique (DFT) is used. The authors must justify the use of the PBE functional.

4-      The authors are obliged to declare the base used along their calculation and why they use this basis set?

5-      The quality of figures are very poor and should be improved to read easily.

6-      The examination of fig 9., show the broadness of the spectrum in the range 3000-3500 cm-1. The authors must improve this broadness. Some papers can help them such as :

7-      DOI:10.1016/j.molstruc.2007.02.040, https://doi.org/10.1021/acs.jpca.9b09655, DOI:10.1016/j.theochem.2007.06.016, https://doi.org/10.1002/9781119165156.ch6. I suggest to the authored to add these refs. Using optimized structural data and vibrational assignments, a comparison with similar structures reported in the literature should be made.

8-      Before submitting to a journal, authors must check for typographical mistakes and references format.
